# Development of Physical Activity Guidelines for a Healthy China Using the Life Cycle Concept: The Perspective of Policy Tools from Five Countries

**Jing Wang** [1], **Fanghui Li** [1,2,]*, **Liang Wu** [2], **Zhuangzhi Wang** [1], **Tian Xie** [1], **Ling Ruan** [3], **Shizhan Yan** [4] **and Yingmin Su** [5,]*

1    School of Sport Sciences, Nanjing Normal University, Nanjing 210023, China
2    School of Sport Sciences, Zhaoqing University, Zhaoqing 526061, China
3    Department of Physical Education, Xi'an Shiyou University, Xi'an 710000, China
4    Postdoctoral Workstation, Jiangsu Research Institute of Sports Science, Nanjing 210023, China
5    School of Sport and Physical Education, North University of China, Taiyuan 030051, China
*    Correspondence: 12356@njnu.edu.cn (F.L.); w12011@njnu.edu.cn (Y.S.)

**Abstract:** Developing physical activity guidelines based on the life cycle concept is conducive to accelerating the realization of the goal of "all-round, full-cycle maintenance and protection to greatly improve people's health" in the Healthy China 2030 Planning Outline. Based on a policy tools perspective, this study uses the text analysis method to collect and analyze physical activity guidelines based on the life cycle concept from five economically developed countries: the USA, Japan, Canada, Australia, and the UK. The policy tools, country data, and stages of the life cycle were used to develop physical activity guidelines in China to accelerate the realization of the Healthy China 2030 strategy based on the following principles: (1) Strengthen sectoral cooperation and establish a system of policy instruments; (2) increase publicity and scientific awareness of physical activity and exercise; (3) focus on talent cultivation and improve guideline research and development; and (4) mobilize the power of all sectors to promote the implementation of physical activity guidelines.

**Keywords:** life cycle; physical activity guidelines; policy tools; Healthy China 2030; public health

## 1. Introduction

At the end of 2019, the outbreak of COVID-19 significantly impacted the lives of millions of people as a global public health emergency and led to an increased focus on the importance of health [1]. According to the WHO (World Health Organization), physical activity is any bodily movement produced by skeletal muscles that expends energy, including activities during work, games, household chores, travel, and recreational activities (e.g., jogging, cycling, dancing, etc.) [2]. The latest research has demonstrated that physical activity plays an important role in countering COVID-19 infection by improving the body's immunity, fighting against the new coronavirus infection, and speeding up the recovery process [3]. In the post-epidemic era, Chinese people's awareness of the importance of daily physical activity has also increased [4]. Additionally, the view of lifelong sports, where physical activity continues through a person's life cycle, has become more prominent in China [5]. The life cycle refers to life from the beginning of the binding of reproductive cells until death, including different stages, and this concept is currently used in many fields of research [6]. China's application of the life cycle in public health began through a document promulgated by the State Council in 2013, namely, Several Opinions of the State Council on Promoting the Development of the Health Service Industry [7]. The life cycle concept is being increasingly used in public health in China. Zhang, a Chinese scholar, divided the life cycle into five stages: fetal, childhood, adolescence, adult, and older life [8]. However, although several plans and policies for physical activity have been

published in China, as this is a relatively new field, a mature system and guidelines for some life cycle stages have not yet been developed. There is also no guideline confirming the life cycle stages. Some economically developed countries have developed internationally recognized guidelines for physical activity guidelines and have also encountered similar social problems as China (e.g., population aging, high obesity rates, etc.). Therefore, the experience of developing physical activity guidelines in these countries will give China inspiration [9]. At present, Chinese researchers focusing on foreign physical activity guidelines have mainly focused on one life cycle stage or examined certain stages between several countries. There has been little research as an overall analysis or on physical activity guidelines at all life cycle stages. Hence, this study examines the physical activity guidelines and relevant policies about physical activity guidelines in five economically developed countries (the USA, Japan, Canada, Australia, and the UK) and builds a three-dimensional framework of policy tools, country data, and the life cycle to summarize the experiences and characteristics of each country. The framework is then used to provide recommendations for developing guidelines for all stages based on the life cycle and guiding people to do physical activity effectively to improve their physical fitness. The results can be used to accelerate the realization of Healthy China 2030.

## 2. Materials and Method

### 2.1. Sample Selection

Physical activity guidelines at all stages of the life cycle in five countries (the USA, Canada, Australia, the UK, and Japan) were collated in two ways. One was through various databases, such as China National Knowledge Infrastructure, PubMed, and Web of Science. The key search words included "physical activity guideline/recommendation/statement for pregnancy/kids/adolescent/adults/the elderly" and "physical activity guideline in the USA/Japan/Canada/Australia/the UK". Another way was from the official government websites, health and education sectors, and associations of the five countries. The selected physical activity guidelines or relevant policies adhered to the following criteria:

(1)    Authority: The selected physical activity guidelines or relevant policies must be published by official authorities, signed by the agency, and accessible on the official website.

(2)    Precision: To ensure the precision of the analysis, the documents we selected included guidelines, principles, activity recommendations, and manuals that involved the content of physical activity recommendations.

Following the screening principles above and due to incomplete statistics, 70 documents were chosen and numbered chronologically according to their publication date (Table 1).

**Table 1.** List of documents on physical activity guidelines in economically developed countries.

| Number | Time | Document | Country |
|---|---|---|---|
| 1 | 1978 | Recommendation on the Quality and Quantity of Physical Exercise to Promote and Maintain Adults' Health | the USA |
| 2 | 1985 | Principles of Guiding Exercise During and after Pregnancy | the USA |
| 3 | 1988 | Opinion Statement on the Physical Fitness of Children and Adolescents | the USA |
| 4 | 1990 | Recommendation on the Quality and Quantity of Physical Exercise to Promote and Maintain Adults' Health | the USA |
| 5 | 1990 | Healthy Citizen 2000 | the USA |
| 6 | 1994 | Exercise Guidelines During and after Pregnancy | the USA |
| 7 | 1995 | Towards the Future—National Sports Standards: Content and Evaluation Guidelines | the USA |
| 8 | 1996 | Physical Activity and Health | the USA |
| 9 | 1997 | Promoting Lifelong Physical Activity for Youth: A Guideline to Community and School Projects | the USA |
| 10 | 1998 | Recommendation on the Quality and Quantity of Physical Exercise to Promote and Maintain Adults' Cardiopulmonary, Muscle Strength, and Flexibility | the USA |
| 11 | 1998 | Physical Activity and Health Promotion for Adults: Evidence and Impact | the UK |
| 12 | 1998 | Physical Activity Guidelines for Active Living in Canada | Canada |
| 13 | 1999 | Health Activity Guidelines and Handbook for Canadian Elderly People | Canada |

**Table 1.** *Cont.*

| Number | Time | Document | Country |
| --- | --- | --- | --- |
| 14 | 1999 | Physical Activity Guidelines for Australian | Australia |
| 15 | 2000 | Basic Plan for Sports Rejuvenation | Japan |
| 16 | 2000 | Healthy Japan 21 | Japan |
| 17 | 2000 | American 3~5 Years Old Children's Sports Suitability Practice Program | the USA |
| 18 | 2000 | Healthy Citizen 2010 | the USA |
| 19 | 2001 | Five-Year Plan to Revitalize Early Childhood Education | Japan |
| 20 | 2002 | Benefits and Risks of Exercise During Pregnancy | Japan |
| 21 | 2002 | Exercise Guidelines during Pregnancy and Postpartum | the USA |
| 22 | 2002 | A Positive Start: Sports Guidelines for Children from 0 to 5 Years Old | the USA |
| 23 | 2002 | Physical Activity Guidelines for Canadian Children and Adolescents | Canada |
| 24 | 2002 | Benefits and Risks of Exercise During Pregnancy | Australia |
| 25 | 2003 | Physical Activity Guidelines for School-age Children | the USA |
| 26 | 2003 | Exercise Guidelines During Pregnancy and Postpartum | Canada |
| 27 | 2004 | At Least 5 Days a Week: About Physical Activity and Its Relationship with Health | the UK |
| 28 | 2006 | Exercise Guidelines for Health Promotion 2006 | Japan |
| 29 | 2006 | Pregnancy Exercise Statement | the UK |
| 30 | 2008 | Physical Activity Guidelines for American | the USA |
| 31 | 2009 | Appropriate Educational Practice Program for American Children Aged 0–8 | the USA |
| 32 | 2009 | Australia's National Activity Recommendations for Seniors | Australia |
| 33 | 2009 | Physical Activity Guidelines for Australians 0–5 Years | Australia |
| 34 | 2010 | Healthy Citizen 2020 | the USA |
| 35 | 2010 | Physical Activity Guidelines for Briton | the UK |
| 36 | 2011 | Children's Obesity Prevention Policy | the USA |
| 37 | 2011 | Action Guidelines to Promote Youth Physical Activity | the USA |
| 38 | 2011 | Sedentary Behavior Guidelines for Canada's Children and Adolescents | Canada |
| 39 | 2011 | Physical Activity Guidelines for Canadians | Canada |
| 40 | 2011 | Physical Activity Guidelines (Under 5) | the UK |
| 41 | 2011 | Start Activity, Stay Active | the UK |
| 42 | 2011 | Report of the Chief Medical Officer of the UK | the UK |
| 43 | 2012 | Basic Plans for Sports | Japan |
| 44 | 2012 | Healthy Japan 21 | Japan |
| 45 | 2012 | Sports Guidelines for Children | Japan |
| 46 | 2012 | Physical Activity Improvement Strategy for American Adolescents | the USA |
| 47 | 2012 | Physical Activity Guidelines for Canada's Children Aged 0–5 | Canada |
| 48 | 2013 | Guidelines for Toddler Sports | Japan |
| 49 | 2013 | Health Promotion Physical Activity Guidelines | Japan |
| 50 | 2013 | Sports During Pregnancy and Postpartum | Australia |
| 51 | 2014 | Physical Activity and Sedentary Guidelines for Australians | Australia |
| 52 | 2014 | Let Us Move: Reduce Sedentary Life and Live a Positive Life (18–24 years old) | Australia |
| 53 | 2014 | Sports During Pregnancy | Australia |
| 54 | 2015 | Exercise Guidelines During Pregnancy and Postpartum | the USA |
| 55 | 2015 | Pregnancy and Exercise | Australia |
| 56 | 2016 | Canada's 24-h Activity Guidelines for Children and Adolescents: Combining Physical Activity, Sedentary Behavior, and Sleep | Canada |
| 57 | 2016 | Sports During Pregnancy and Postpartum | Australia |
| 58 | 2017 | Guide to Obstetrics and Gynecology Diagnosis and Treatment | Japan |
| 59 | 2017 | 24-h Exercise Guidelines for Canada's Toddlers | Canada |
| 60 | 2017 | Physical Activity Guidelines for Australian Children (0–5 Years Old) | Australia |
| 61 | 2018 | Physical Activity Guidelines for Americans | the USA |
| 62 | 2018 | Physical Activity Guidelines During Pregnancy in Canada | Canada |
| 63 | 2018 | Canada's Shared Vision of Increasing Physical Activity/Decreasing Sedentary Activity: Let Us Exercise | Canada |
| 64 | 2018 | Exercise During Pregnancy and Postpartum | Australia |
| 65 | 2019 | Australian 24-h Activity Guidelines for Children and Adolescents: Integrating Physical Activity, Sedentary Behavior and Sleep | Australia |
| 66 | 2019 | Physical Activity Guidelines for Britons | the UK |
| 67 | 2020 | Exercise Guidelines During Pregnancy and Postpartum | the USA |
| 68 | 2020 | Physical Activity Guidelines: Report of the Chief Medical Officer of the UK | the UK |
| 69 | 2021 | Physical Activity Guidelines for 18–64 Years Old Australian Adults | Australia |
| 70 | 2021 | Physical Activity and Exercise Guidelines During Pregnancy | Australia |

### 2.2. Three-Dimensional Framework Construction

A three-dimensional framework was constructed for the analysis. The X-dimension is policy tools, which are used to implement and bridge policy objectives and results [10]. There are many classifications for policy tools. The policies and documents collected in this paper were divided into supplied, environmental, and demand policy tools according to the

different effects of policy influence developed by Rothwell and Zegveld [11]. The supplied policy tools mainly provide information, consulting, and education services at different life cycle stages, such as physical activity guidelines and brochures that motivate people to undertake physical activity [12]. Environmental policy tools refer to governmental strategic measures, such as management plans, laws, and regulations, and they have a persistent and indirect effect on people's physical activity. Demand policy tools refer to documents promulgated by the government to stimulate people to take the initiative to do physical activity. For example, relevant reports and statements on the health benefits of exercise motivate people from all life cycle stages to do physical activity [13].

The Y-dimension is the countries chosen, and recognized and long-lasting physical activity guidelines and relevant policies were chosen from the USA, Canada, and Australia. Additionally, the UK and Japan were included because, like China, they are in a "post-Olympic" period. In addition, the five stages (fetal, children, adolescence, adult, and older) of the life cycle are the Z-dimension. According to the above classification criteria, a three-dimensional diagram is illustrated in Figure 1.

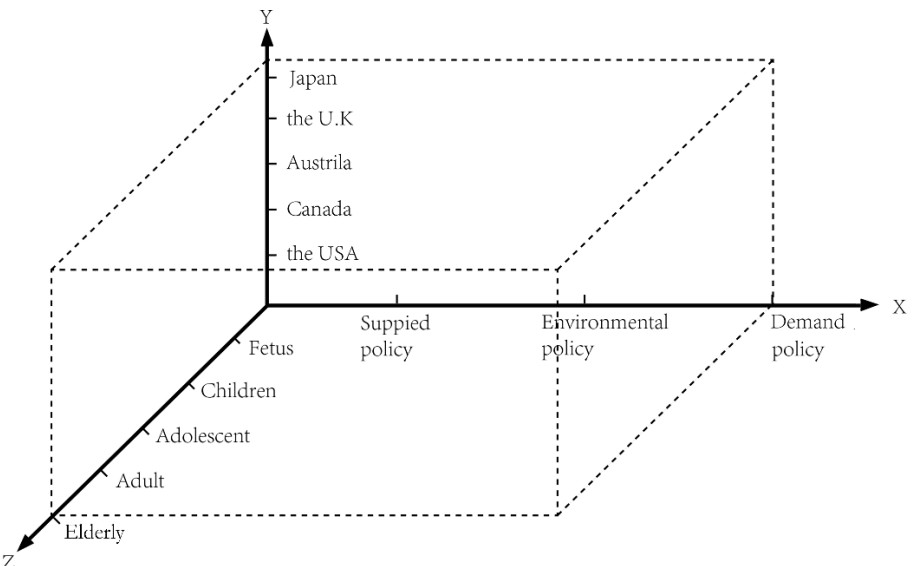

**Figure 1.** Diagram chart of three-dimensional framework.

*2.3. Sample Encoding*

The coding process of this paper went through three rounds with five participants. In the first round, after a preliminary reading of the selected policies, two people classified the kinds of policy tools to determine the coding of the X-dimension. After completion, the reviewers exchanged the encoding results with each other and invited a third person to conduct a second round to discuss the differences between the results and determine the final version. The third round involved another two reviewers identifying the life cycle stages in every document we collected to determine the code of the Z-dimension. The coding table was not modified further after everyone agreed, and the author analyzed the results. The specific coding of each dimension is as follows: in the X-dimension, supplied policy tools were coded as 01, environmental policy tools as 02, and demand policy tools as 03. In the Y-dimension, the USA was coded as US, Canada as CN, Australia as AU, the UK as UK, and Japan as JP. In the Z-dimension, fetal was coded as 01, childhood as 02, adolescence as 03, adult as 04, and older life as 05. If a document involved multiple stages, it was coded in both stages. The encoding results are illustrated in Table 2.

**Table 2.** Encoding table of documents for physical activity guidelines in developed countries.

| Number | Time | Document | Code |
|---|---|---|---|
| 1 | 1978 | Recommendation on the Quality and Quantity of Physical Exercise to Promote and Maintain Adults' Health | 01-US-04 |
| 2 | 1985 | Principles of Guiding Exercise During and after Pregnancy | 01-US-01 |
| 3 | 1988 | Opinion Statement on the Physical Fitness of Children and Adolescents | 03-US-0203 |
| 4 | 1990 | Recommendation on the Quality and Quantity of Physical Exercise to Promote and Maintain Adults' Health | 01-US-04 |
| 5 | 1990 | Healthy Citizen 2000 | 02-US-01:05 |
| 6 | 1994 | Exercise Guidelines During and after Pregnancy | 01-US-01 |
| 7 | 1995 | Towards the Future—National Sports Standards: Content and Evaluation Guidelines | 01-US-02 |
| 8 | 1996 | Physical Activity and Health | 03-US-0203 |
| 9 | 1997 | Promoting Lifelong Physical Activity for Youth: A Guideline to Community and School Projects | 01-US-03 |
| 10 | 1998 | Recommendation on the Quality and Quantity of Physical Exercise to Promote and Maintain Adults' Cardiopulmonary, Muscle Strength, and Flexibility | 01-US-04 |
| 11 | 1998 | Physical Activity and Health Promotion for Adults: Evidence and Impact | 03-UK-04 |
| 12 | 1998 | Physical Activity Guidelines for Active Living in Canada | 01-CN-04 |
| 13 | 1999 | Health Activity Guidelines and Handbook for Canadian Elderly People | 01-AU-05 |
| 14 | 1999 | Physical Activity Guidelines for Australian | 01-AU-04 |
| 15 | 2000 | Basic Plan for Sports Rejuvenation | 02-JP-02:05 |
| 16 | 2000 | Healthy Japan 21 | 02-JP-04:05 |
| 17 | 2000 | American 3~5 Years Old Children's Sports Suitability Practice Program | 01-US-02 |
| 18 | 2000 | Healthy Citizen 2010 | 02-US-01:05 |
| 19 | 2001 | Five-Year Plan to Revitalize Early Childhood Education | 02-JP-02 |
| 20 | 2002 | Benefits and Risks of Exercise During Pregnancy | 03-JP-01 |
| 21 | 2002 | Exercise Guidelines during Pregnancy and Postpartum | 01-US-01 |
| 22 | 2002 | A Positive Start: Sports Guidelines for Children from 0 to 5 Years Old | 01-US-02 |
| 23 | 2002 | Physical Activity Guidelines for Canadian Children and Adolescents | 01-CN-0203 |
| 24 | 2002 | Benefits and Risks of Exercise During Pregnancy | 03-AU-01 |
| 25 | 2003 | Physical Activity Guidelines for School-age Children | 01-US-02 |
| 26 | 2003 | Exercise Guidelines During Pregnancy and Postpartum | 01-CN-01 |
| 27 | 2004 | At Least 5 Days a Week: About Physical Activity and Its Relationship with Health | 03-UK-0203 |
| 28 | 2006 | Exercise Guidelines for Health Promotion 2006 | 01-JP-04 |
| 29 | 2006 | Pregnancy Exercise Statement | 03-UK-01 |
| 30 | 2008 | Physical Activity Guidelines for American | 01-US-01:05 |
| 31 | 2009 | Appropriate Educational Practice Program for American Children Aged 0–8 | 01-US-02 |
| 32 | 2009 | Australia's National Activity Recommendations for Seniors | 01-AU-05 |
| 33 | 2009 | Physical Activity Guidelines for Australians 0–5 Years | 01-AU-02 |
| 34 | 2010 | Healthy Citizen 2020 | 02-US-01:05 |
| 35 | 2010 | Physical Activity Guidelines for Briton | 01-UK-01:05 |
| 36 | 2011 | Children's Obesity Prevention Policy | 02-US-02 |
| 37 | 2011 | Action Guidelines to Promote Youth Physical Activity | 01-US-03 |
| 38 | 2011 | Sedentary Behavior Guidelines for Canada's Children and Adolescents | 01-CN-0203 |
| 39 | 2011 | Physical Activity Guidelines for Canadians | 01-CN-01:05 |
| 40 | 2011 | Physical Activity Guidelines (Under 5) | 01-UK-02 |
| 41 | 2011 | Start Activity, Stay Active | 02-AU-0203 |
| 42 | 2011 | Report of the Chief Medical Officer of the UK | 03-UK-01:05 |
| 43 | 2012 | Basic Plans for Sports | 02-JP-02:05 |
| 44 | 2012 | Healthy Japan 21 | 02-JP-0405 |
| 45 | 2012 | Sports Guidelines for Children | 01-JP-02 |
| 46 | 2012 | Physical Activity Improvement Strategy for American Adolescents | 02-US-03 |
| 47 | 2012 | Physical Activity Guidelines for Canada's Children Aged 0–5 | 01-CN-02 |
| 48 | 2013 | Guidelines for Toddler Sports | 01-JP-02 |
| 49 | 2013 | Health Promotion Physical Activity Guidelines | 01-JP-02:05 |
| 50 | 2013 | Sports During Pregnancy and Postpartum | 03-AU-01 |
| 51 | 2014 | Physical Activity and Sedentary Guidelines for Australians | 01-AU-01:05 |
| 52 | 2014 | Let Us Move: Reduce Sedentary Life and Live a Positive Life (18–24 years old) | 02-AU-04 |
| 53 | 2014 | Sports During Pregnancy | 03-AU-01 |
| 54 | 2015 | Exercise Guidelines During Pregnancy and Postpartum | 01-US-01 |

**Table 2.** *Cont.*

| Number | Time | Document | Code |
|---|---|---|---|
| 55 | 2015 | Pregnancy and Exercise | 03-AU-01 |
| 56 | 2016 | Canada's 24-h Activity Guidelines for Children and Adolescents: Combining Physical Activity, Sedentary Behavior, and Sleep | 01-CN-0203 |
| 57 | 2016 | Sports During Pregnancy and Postpartum | 03-AU-01 |
| 58 | 2017 | Guide to Obstetrics and Gynecology Diagnosis and Treatment | 01-JP-01 |
| 59 | 2017 | 24-h Exercise Guidelines for Canada's Toddlers | 01-CN-01 |
| 60 | 2017 | Physical Activity Guidelines for Australian Children (0–5 Years Old) | 01-AU-02 |
| 61 | 2018 | Physical Activity Guidelines for Americans | 01-US-01:05 |
| 62 | 2018 | Physical Activity Guidelines During Pregnancy in Canada | 01-CN-01 |
| 63 | 2018 | Canada's Shared Vision of Increasing Physical Activity/Decreasing Sedentary Activity: Let Us Exercise | 02-CN-01:05 |
| 64 | 2018 | Exercise During Pregnancy and Postpartum | 03-AU-01 |
| 65 | 2019 | Australian 24-h Activity Guidelines for Children and Adolescents: Integrating Physical Activity, Sedentary Behavior and Sleep | 01-AU-0203 |
| 66 | 2019 | Physical Activity Guidelines for Britons | 01-UK-01:05 |
| 67 | 2020 | Exercise Guidelines During Pregnancy and Postpartum | 01-US-01 |
| 68 | 2020 | Physical Activity Guidelines: Report of the Chief Medical Officer of the UK | 01-UK-01:05 |
| 69 | 2021 | Physical Activity Guidelines for 18–64 Years Old Australian Adults | 01-UK-04 |
| 70 | 2021 | Physical Activity and Exercise Guidelines During Pregnancy | 01-AU-01 |

## 3. Results

### 3.1. X-Dimension Analysis

Supplied policy tools were the most used tool in all five countries, which accounted for 62.8% of the total. Most of the supplied policy tools were physical activity guidelines developed jointly by experts in multidisciplinary fields based on scientific evidence. Their contents are specific, refined, and cover the type, time, and intensity of physical activity and, therefore, have strong operability and provide specific guidance to people from all life cycle stages on how to do physical activity effectively [14]. They are important policies for guiding and promoting physical activity. Environmental policy tools were used less frequently, accounting for 18.6%, and most were national strategic plans, such as Healthy People from the USA or the Basic Plan for Sports Rejuvenation from Japan. Although physical activity guidelines were only part of these policies, they covered almost all life cycle stages. These policies indicate that the guidelines from the macro-policy environment are national strategies that promote people's health as an important index affecting how people carry out daily physical activity. Supplied policy tools were used relatively less, accounting for 18.6%. Researchers have demonstrated that one of the reasons people do not actively engage in physical activity is that there is limited awareness of the benefits of physical activity [15]. The publication of the Declaration on the Benefits of Pregnancy, Physical Activity, and Health and other demand policies can strengthen people's awareness of all life cycle stages, encourage them to do physical activity to promote their health, and improve the subjective initiative of doing daily physical activity.

### 3.2. Y-Dimension Analysis

The Y-dimension is an overview of the five countries' physical activity guidelines throughout the life cycle. The United States is a pioneer in developing and practicing the guidelines as it has issued a total of 24 documents, accounting for 34.3%. It was also the first country to publish physical activity guidelines for every life cycle stage. In 1985, the American College of Obstetricians and Gynecologists (ACOG) published the first set of guidelines for pregnancy, Guidelines for Pregnancy and Postpartum; in doing so, it became the first country to develop and implement guidelines for physical activities and pregnancy [16]. In 2008, the US Department of Health and Human Services published the Physical Activity Guidelines for Americans, the first physical activity guidelines providing classification guidance for people at different life cycle stages [17]. The Physical Activity

Guidelines for Americans publication included the "dose–effect" relationship based on scientific research results from sports and health and recommended scales for physical activities at different life cycle stages as a precaution. The development of physical activity guidelines for over 40 years in the USA has been a systematic, scientific, and continuous transition from medical and clinical rehabilitation to public health [18,19].

Canada also has extensive experience in developing physical activity guidelines. From 1998–2003, guidelines for adulthood, older adults, childhood, adolescence, and fetal life were developed successively [20]. In 2006, a guideline covering all stages of people's life cycle was prepared and published as the Canadian Physical Activity Guidelines [21]. Accompanying this, the publication of Physical Activity and Sedentary Behavior Guidelines for Canadian Children and Youth raised concern, and it was also the first physical activity guideline to address sedentary behavior in adolescents [22]. Due to the lack of physical activity and sleep time among children and adolescents in Canada, the government issued the Canadian 24-h Movement Guidelines for Children and Youth: Integrating Physical Activity, Sedentary Behavior, and Sleep to raise awareness of the importance of a 24-h combination of behavior (physical activity, sedentary behavior, and sleep) for the physical and mental health of children and adolescents. It subsequently became the model for developing guidelines in other countries [23].

In 1999, the Australian government published the first edition of the Physical Activity Guidelines for Australians to reverse the trend of obesity, but this was only for adults [24]. In 2014, to promote the overall health of Australians, the Australian Department of Health and Welfare, in conjunction with the Australian Medical Association and others, drew on the successful experiences in the USA and Canada and, in accordance with the recommendations of the World Health Organization, developed the Australian Physical Activity and Sedentary Guidelines, which covers all life cycle stages [25]. The different sections were then supplemented and improved (e.g., 0–5 years old, 6–17 years old, 18–64 years old, 64 years old, or special populations), reflecting the continuity and advancement of the guidelines.

The UK government implements health promotion programs based on physical activity. After the successful bid for the 2005 London Olympic Games, the UK government took this opportunity to promulgate a series of sports policies to improve overall physical fitness and the international competitiveness of national sports. Start active, stay active: report on physical activity in the UK, the first physical activity guidelines covering all life cycle stages, were published in 2011 to promote public health through physical activities; the guidelines have been continuously updated [26].

Japan was the first Asian country to develop physical activity guidelines. The Japanese government has promulgated a series of policies and regulations, such as the Basic Law on Sports and the Basic Plan for Sports Rejuvenation, that provides the guidelines with legal status to ensure their effective implementation [10]. Japan's aging population is a severe problem; the aging rate was 7% in 1970 and is expected to reach 32.3% by 2050 [27]. Consequently, the Japanese government has attached great importance to solving the country's aging problem. Each edition of Healthy Japan 21 sets certain daily physical activity goals for older adults. Although Japan's physical activity guidelines do not cover all life cycle stages, the specific methods for evaluating the amount of physical activity are linked to daily living and are more operational [27,28].

The development processes for physical activity guidelines based on life cycle stages in the countries mentioned above include the initial exploration of mature systems. However, due to the differences in politics, economy, culture, and education, each country has focused on a particular life cycle stage when developing guidelines. Additionally, each country has several effective guidelines with specific characteristics and a high degree of international recognition; for example, the Physical Activity Guidelines for Americans and the Canadian 24-h Movement Guidelines for Children and Youth: Integrating Physical Activity, Sedentary Behavior, and Sleep are examples that other countries can use in developing similar guidelines.

*3.3. Z-Dimension Analysis*

The number of guidelines published at each stage varies among the countries because of the differences between their historical and cultural backgrounds, basic national conditions, and social problems. Since the end of the 20th century, a lack of physical activity, long sedentary lifestyles, and other negative behaviors have increased obesity among children and adolescents, especially in certain economically developed countries [29]. Eight national surveys conducted in the USA and Canada between 1972 and 1983 have demonstrated a lack of physical activity among children and adolescents, which prompted the US and Canadian governments to accelerate the development of guidelines for physical activity among children and adolescents [30,31]. Since the success of the 2005 London Olympic bid, the UK government has focused on the future development of sports in the country to create competitive sports reserves, personnel, and strategic planning [32]. Children and adolescent sports have become the most important area for competitive sports as this is where the reserve talent is trained. For these reasons, governments in the USA, Canada, and the UK have issued many guidelines on physical activity in childhood and adolescence (Figure 2). Japan's aging rate has been at the highest level globally, which has led to labor shortages, rising medical costs, and other issues, creating a burden on families and society [33]. Consequently, the Japanese government issued a series of relevant policy documents to improve the overall health of the country's older adults to ease the social burden [34].

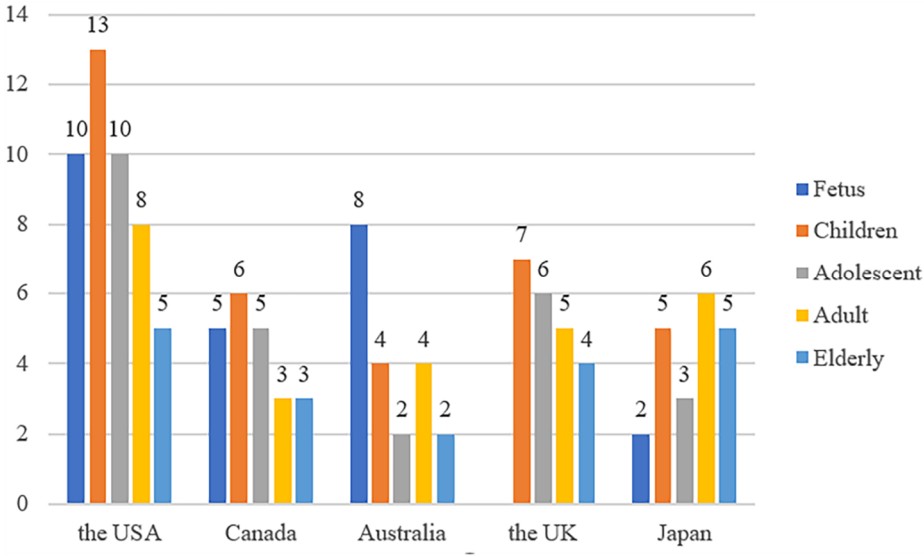

**Figure 2.** Percentage chart of the number of guidelines published at every stage in each country.

The guidelines for physical activity at all stages of the life cycle for the countries mentioned above are as follows:

(1) Fetus: At present, the recommended amount of physical activity during pregnancy is at least 150 min of moderate-intensity aerobic activity and muscle-strengthening physical activity per week [35]. Canada and Australia have also suggested that pregnant people need to do daily pelvic-muscle-strengthening physical activities (Table 3). While developing the guidelines, these countries have focused on scientific theory and clinical medical recommendations to combine the multidisciplinary knowledge of physiology and sports and provide a full range of multi-level comprehensive physical activity advice for pregnant people. The safety of pregnant people is also a key consideration in the development process of the guidelines. The content of physical activity programs has been evaluated for safety and risk assessments by experts and makes it clear that pregnant people should stop exercising in some conditions [36–39].

(2)    Children and adolescents: After several updates and improvements, the recommendations on the time, intensity, frequency, and type of physical activity for children and adolescents in these countries gradually became the same. They all suggest at least 60 min of moderate to intense aerobic physical activity per day and at least three musculoskeletal strengthening physical activities per week (Table 3). This matches the WHO's current recommendation for children and adolescents [40]. Except for physical activities, the Canadian 24-h Movement Guidelines for Children and Youth: Integrating Physical Activity, Sedentary Behavior, and Sleep has been a model for other countries. It highlights the synergies between physical activity, sedentary time, and sleep and calls on parents, teachers, and stakeholders to build a healthy 24-h lifestyle for children and adolescents [41]. Although most physical activity guidelines for adolescents have been developed and published as combined guidelines for children, some countries have published separate guidelines on adolescence [41,42]. In the USA, the Physical Activity Guidelines for Americans: Strategies to Increase Physical Activity Among Youth have emphasized the need to promote the construction of physical activity sites for adolescents, with the greatest emphasis on schools, followed by families and communities [43]. This creates the linked effect of "school—family—community", which is led by physical education at school, involving parental physical activity guidance and community-provided environmental support. This adolescent physical health model has been recognized and promoted in many other countries as well.

(3)    Adults: 150–300 min of MVPA (moderate-to-vigorous physical activity) per week, 75–150 min of vigorous physical activity, or an equivalent combination of both is the current recommendation for adults in these countries' physical activity guidelines (Table 3). In addition, the current recommendations state that adults should do muscle-strengthening activities involving major muscle groups at least two times per week. All countries promote the "comprehensive health concept" lifestyle, including physical activity, sedentary time, and sleep. As adults have a wide range of workplaces, in the guidelines, walking, stair climbing, cycling, housework, and other more life-oriented sports are recommended. It is also recommended that individuals make appropriate adjustments according to their work situation to achieve physical activity goals.

(4)    Older adults: The total amount of physical activity recommended for older adults is consistent with that of adults in the guidelines of these countries. However, the recommended types of physical activity programs are more tailored to the needs of older people. As individuals age, their muscle strength and balance decrease, which increases the risk of falls; hence, the types of physical activity recommended for older adults in the guidelines are mainly to improve muscle strength and balance. As it is risky for older adults to do physical activity at a high intensity by themselves, the guidelines recommend lighter exercises such as walking and social dances to avoid safety problems [44,45].

It can be found from the above guidelines that muscle-strengthening exercise, which is a voluntary activity that includes the use of weight machines, hand-held weights, or own body weight and has multiple and unique health benefits for people, plays an important role in the physical activity guidelines of all countries mentioned above [46]. Studies have shown that compared to engaging in either the muscle-strengthening exercise guideline ($\geq$2 sessions/week) or the aerobic MVPA guideline alone ($\geq$150 min/week), the combination of both may be most beneficial for the prevention and/or management of multiple prevalent chronic health conditions [47,48]. In addition, sedentary time is also highlighted in the guidelines, and there are many practical tips to reduce sedentary behavior in daily life or at work, such as walking around when talking on your mobile phone or asking your boss for a "walk and talk" meeting rather than a sitting meeting [49]. A consensus has been reached on the fact that a small amount of activity is better than inactivity. Some studies have suggested that bouts of physical activity as short as 10 min are associated with similar

health benefits to physical activity accumulated in longer bouts [50]. Moreover, there are four principal domains in which PA can be performed: leisure, work, transportation, and domestic life, which have been shown to display independent associations with health outcomes [51]. It is essential to recommend suitable exercises for different domains. For example, the Australian physical activity guidelines not only state the standards of exercise duration but also have different suggestions for different domains, such as "build activity into your day", "active at work", "active indoors", and so on [49].

**Table 3.** The amount of physical activity for all stages of the life cycle in economically developed countries.

|  | The USA | Canada | Australia | The UK | Japan |
|---|---|---|---|---|---|
| Fetus | at least 150 min of moderate-intensity aerobic and muscle-strengthening exercises per week. | at least 150 min of moderate-intensity aerobic and muscle-strengthening exercises per week; pelvic-muscle-strengthening exercises per day. | at least 150–300 min of moderate-intensity or 75–150 min of vigorous-intensity aerobic physical activity and muscle-strengthening exercises per week; pelvic-muscle-strengthening exercises per day. | at least 150 min of moderate-intensity aerobic and muscle-strengthening exercises per week. | at least 150 min of moderate-intensity aerobic exercise per week. |
| Children and adolescent | at least 60 min of moderate-to-vigorous physical activity per day; at least 3 days muscle-strengthening and bone-strengthening physical activity per week. | at least 60 min of moderate-to-vigorous physical activity per day; at least 3 days muscle-strengthening physical activity per week. | at least 60 min of moderate-to-vigorous aerobic physical activity per day; at least 3 days muscle-strengthening exercise per week. | at least 60 mins per day; engage in a variety of types and intensities of physical activity to develop movement skills, muscular fitness, and bone strength. | at least 60 min of moderate-to-vigorous physical activity per day; at least 3 days muscle-strengthening physical activity per week. |
| Adult | 150–300 min of MVPA per week, 75–150 min of vigorous physical activity, or an equivalent combination of both; at least 2 days of muscle-strengthening per week. | at least 150 min of MVPA per week; at least 2 days of muscle-strengthening activities per week. | at least 150–300 min of moderate-intensity or 75–150 min of vigorous-intensity aerobic physical activity per week; at least 2 days of muscle-strengthening activities per week. | at least 150 min of moderate intensity activity or 75 min of vigorous-intensity activity, or even shorter durations of very vigorous intensity activity per week; do muscle-strengthening activities at least two days per week. | at least 150 min of moderate-to-vigorous physical activity per week. |
| Elderly | 150–300 min of MVPA per week, 75–150 min of vigorous physical activity, or an equivalent combination of both; at least 2 days of muscle-strengthening per week. | at least 150 min of moderate-intensity aerobic physical activity per week; at least 2 days a week of musculoskeletal intensive exercise and balance training. | at least 30 min of moderate-intensity aerobic physical activity per day. | at least 150 min of moderate intensity activity or 75 min of vigorous-intensity activity; improving or maintaining muscle strength, balance, and flexibility at least two days per week. | at least 150 min of moderate-to-vigorous physical activity per week; at least 2 days a week of muscle-strengthening activities and balance training. |

## 4. Analysis of the Situation in China

### 4.1. The Development Status of China's Physical Activity Guidelines

After the founding of the People's Republic of China, the government specifically proposed implementing a "whole nation system" strategy, concentrating on improving the comprehensive strength of competitive sports. After the reform and opening-up of China, the positioning of Chinese sports was transformed from "sports saving the country" to "sports power". After the 2008 Beijing Olympic Games, the positioning of "sports power" began to gradually transition to "sports giant". In 2019, China officially promulgated the policy of the "Outline for Building a Leading Sports Nation" to build a modern socialist country by 2050. In the evolution of history, sports have played an increasingly important role in China. Meanwhile, the public's awareness of physical activity is also becoming increasingly profound. According to the 2020 survey of the National Fitness Bulletin, while the proportion of people participating in physical activity in China continues to grow,

there is a lack of physical activity guidelines in China, which has led to a lack of effective guidance for people to participate in physical activity.

### 4.1.1. Lacking Guidelines for Fetal and Older Adult Life Stages

Before the 21st century, pregnant people were advised to rest and undertake low levels of physical activity due to the influence of traditional concepts. This has led to a weak awareness of the need for physical activity during pregnancy in China [52]. In other countries, exercising during pregnancy has been promoted by publishing physical activity guidelines and organizing sports training activities for pregnant women; hence, Chinese scholars have increasingly demonstrated the importance of physical activity during pregnancy. The increasing prevalence of obesity, hypertension, and diabetes in pregnant women in China has also increased the promotion of pregnancy physical activity [53]. However, no national guidelines have been developed, and the National Fitness Plan documents do not include physical activity for pregnant people.

China has also become an aging society, which creates unique problems. According to the results of the seventh census, the number of people aged 60 years and older in China is 264 million, accounting for 18.70% of the population [54]. Improving the physical activity levels of older adults by developing a healthy lifestyle and healthy aging is the current focus of attention for the Chinese government [55]. The government has also promulgated a series of policies to promote the health of older adults. The results of previous national physical fitness monitoring surveys have demonstrated that the number of older adults who often participate in physical activity in China has been increasing, indicating that the relevant policies have worked [56]. However, one problem that cannot be ignored in the physical monitoring results is that, although regular physical activity has increased, the physical fitness compliance rate has not significantly improved or demonstrated a downward trend. This indicates that some problems need to be addressed by developing physical activity guidelines for older adults, as the elderly are a special group. In the latest Physical Activity Guidelines for the Chinese (2021), the content on how older people should engage in physical activity is not specific, and there is a lack of special physical activity guidelines for the elderly in China [57].

### 4.1.2. Inadequate Guidelines for Children, Adolescents, and Adults

China has attached great importance to the healthy development of children and adolescents and has developed a series of policies and regulations (Table 4). Since the reform and opening up, although some health indicators (e.g., height and physical fitness) in children and adolescents have improved significantly, they are still problems with cardiopulmonary function, endurance quality, decreased average vision, and rising obesity rates [58]. In 2018, the Physical Activity Guidelines for Children and Adolescents in China was published with clear regulations on the intensity, time, and type of physical activity, in line with the five aforementioned countries. However, the guidelines do not provide specific sedentary and sleep time limits and do not emphasize building a "school–family–community" network of physical activity promotion for children and adolescents.

In early 2011, the trial edition of the Physical Activity Guidelines for Chinese Adults was published. However, there was a lack of publicity and promotion, limiting the effect. Additionally, similar to the National Fitness Plan and the Healthy China 2030 Program Outline, the content was not specific and included overall strategies and plans that lacked specific operability [59]. Therefore, the health status of adults in China has not significantly improved, especially for women and rural and low-income people.

**Table 4.** List of guidelines for physical activity at all life stages in China.

| Stages | Time | Document |
|---|---|---|
| Children | 1982 | Notice of the Ministry of Education on the Daily Physical Activity of Primary and Secondary School Students |
| | 1984 | Notice on the Further Development of Sport and Physical Education |
| | 1996 | Charter of Work of Kindergartens |
| | 2002 | Full-time Compulsory Education General High School Physical Education (grades 1–6); Physical Education and Health (grades 7–12) Curriculum Standards |
| | 2005 | Opinions of the Ministry of Education on the Implementation of the Daily Activities of Primary and Secondary School Students |
| | 2011 | National Fitness Program (2011–2015) |
| | 2016 | National Fitness Program (2016–2020) |
| | 2016 | Healthy China 2030 Planning Outline |
| | 2018 | Exercise Guideline for Preschool Children (3–6 Years) (Expert Consensus Edition) |
| | 2018 | Physical Activity Guideline for Chinese Children and Adolescents |
| | 2021 | Physical Activity Guidelines for the Chinese (2021) |
| Adolescent | 2000 | Sports Reform and Outline 2001–2010 |
| | 2007 | Opinions of the State Council of the CPC Central Committee on Strengthening Youth Physical Education to Enhance Youth Physical Fitness |
| | 2011 | National Fitness Program (2011–2015) |
| | 2016 | National Fitness Program (2016–2020) |
| | 2016 | Healthy China 2030 Planning Outline |
| | 2018 | Physical Activity Guideline for Chinese Children and Adolescents |
| | 2020 | Opinions on Deepening the Integration of Physical Education and Promoting the Healthy Development of Adolescents |
| | 2021 | Physical Activity Guidelines for the Chinese (2021) |
| Adult | 2011 | Physical Activity Guideline for Chinese Adults |
| | 2011 | National Fitness Program (2011–2015) |
| | 2016 | National Fitness Program (2016–2020) |
| | 2016 | Healthy China 2030 Planning Outline |
| | 2021 | Physical Activity Guidelines for the Chinese (2021) |
| Elderly | 1999 | Notice on Strengthening Sports for the Elderly |
| | 2000 | Sports Development Plan for the Elderly |
| | 2011 | National Fitness Program (2011–2015) |
| | 2014 | Notice on the Issuance of Core Information on Old Age Health |
| | 2016 | National Fitness Program (2016–2020) |
| | 2016 | Healthy China 2030 Planning Outline |
| | 2016 | Opinions on Further Strengthening Sports Work for the Elderly Under the New Situation |
| | 2021 | Physical Activity Guidelines for the Chinese (2021) |

### 4.1.3. Incomplete System of Policy Tools

From Table 4, it can be seen that most of the current policies issued in China on physical activity guidelines at all life cycle stages are environmental policy tools, and there is a lack of supplied and demand policy tools. Environmental policy tools are from the macro-policy environment and make recommendations for detailed measures but lack the direct push-and-pull effect for people at all life cycle stages to carry out physical activities. At present, China has not yet developed tools such as the Physical Activity Guidelines for Americans and the Canadian Physical Activity Guidelines, which cover all life cycle stages.

### 4.2. Future Development Strategy in China

### 4.2.1. Develop Physical Activity Guidelines for the Fetus and Older Adults

While physical activity is crucial for the fetal and older adult life cycle stages, physical activity guidelines for these life cycle stages have not been developed in China. At the beginning of the life cycle, the fetus stage forms an important foundation for the development of the individual [59]. Researchers have demonstrated that effective physical activity during pregnancy can improve the cardiovascular health of the fetus, prevent hypoglycemia, low birth weight, and congenital malformations in newborns, and improve the development of

the baby's mental and language skills later in life [60,61]. The older adult stage is the last stage of the life cycle. With the body's aging, the most obvious manifestation is the gradual decline of the individual's various body functions, such as motor ability, muscle strength, and cardiovascular function. Older adults undertaking appropriate physical activities can delay physical function decline and prevent and treat various chronic diseases [62].

To develop physical activity guidelines for the fetus stage, first, the traditional concept of pregnant people in China needs to change, and the awareness of the benefits of physical activity during pregnancy needs to increase. Second, a group of experts should establish a system of evidence-based guidelines and provide specific physical activity prescriptions based on the physiological and anatomical characteristics of pregnancy and the health status and needs of pregnant people. Finally, the application coverage of the guidelines needs to be expanded to include the broad range of needs of pregnant people according to their identity, age, gestation period, and environment and provide targeted physical activity prescriptions.

The duration and intensity of physical activity recommendations in the physical activity guidelines of the governments of the USA, Canada, Japan, Australia, and the UK tend to be consistent for older adults. Therefore, they can be used as the main reference standards for national guidelines in China in order to make appropriate adjustments in accordance with the physical condition of older people. Moreover, the guidelines should focus on developing and maintaining muscle strength and balance in older people [63]. Recommended physical activities include tai chi, eight pieces of brocade, square dancing, and other sports, in line with China's traditional culture and older people's lifestyles. Finally, researchers have demonstrated that sedentary behavior has become a new risk factor associated with obesity, diabetes, and cardiovascular disease, especially in older populations [64]. As older people have lower cardiopulmonary fitness and are more susceptible to sedentary behaviors, emphasis should also be placed on limiting their sedentary time.

### 4.2.2. Improve Physical Activity Guidelines in Childhood, Adolescence, and Adulthood

China has now issued the Exercise Guideline for Preschool Children (3–6 Years) (Expert Consensus Edition) and the Physical Activity Guideline for Chinese Children and Adolescents, which provide preliminary physical activity guidelines for children and adolescents. However, the basic framework for a 24-h guide, including physical activity, sedentary time, and sleep, has not yet been established [65]. The physical activity guidelines for children and adolescents in economically developed countries should be used as a reference, with appropriate adjustments. First, the government should understand the overall physical conditions and different health level requirements according to the physical monitoring results of children and adolescents in China and recommend the required physical activity time and intensity, specific sedentary time limits, and appropriate sleep time. According to China's historical culture, geographical environment, and other characteristics, sports activities that stimulate children's and young people's interest in sports and have "elements of life" should be included. Furthermore, the places for physical activity should be identified, and the linkage effect of "school–family–community" should be established. Finally, this should be supplemented with physical activity risks and risk management information.

For adults' physical activities, China promulgated the Physical Activity Guidelines for Chinese Adults (Trial Edition) publication in 2011 [59], but both the publicity efforts and the practicality of the content need to be improved. First, the operability of the recommended content needs to be improved. The recommended physical activity level in the previous version is the thousand-step equivalent, which is metabolically equivalent (MET) to daily or weekly cumulative physical activity time; the recommended strength of the MET should be changed to the highest heart rate percentage, which is easier to understand. Next, strength training should be actively promoted. Finally, the guidelines

should also emphasize the provision of a specific environment and improving facilities for adults to carry out physical activities.

### 4.2.3. Improve the Policy Tools and Develop Guidelines Covering All Stages of the Life Cycle

Most of China's current physical activity guidelines are environmental policy tools, which are macro-focused and lack content and operability. As mentioned above, supplied policy tools provide information, consulting, and education services at different life cycle stages, such as physical activity guidelines and brochures that motivate people to undertake physical activity [12]. Demand policy tools refer to documents promulgated by the government to stimulate people to take the initiative to do physical activity, and both will provide specific and effective guidance on physical activity. Therefore, the policy tool system should be improved, and relevant supplied and demand policy tools need to be developed to achieve a balance. This should draw on the five countries' physical activity guidelines to ensure that guidelines covering all stages of the life cycle are developed for effective guidance to improve the physical fitness of Chinese citizens. For example, China can learn from the recommendations of other countries on the levels of physical activity at all stages of the life cycle and make adjustments based on Chinese physical test report data. In addition, the content of each stage should be as specific as possible. Specific physical activity programs can also be suggested according to the cultural characteristics of China, such as tai chi, square dancing, and eight pieces of brocade.

## 5. Accelerating the Realization of the "Health China 2030" Strategy

### 5.1. Strengthen Sectoral Cooperation and Establish a System of Policy Instruments

Policy tools as a system of supplied, environmental, and demand policy tools have advantages and roles in promoting physical activity; however, when selecting and implementing these policies, the focus should be on supplementing each policy and forming a reasonable layout. It is more important to strengthen the close cooperation between government departments and social organizations to ensure effective policy integration among organizational sectors such as the Ministry of Health, sports bureaus, hospitals, and research institutes and to form a top-down leadership mechanism and a bottom-up feedback mechanism.

### 5.2. Increase Publicity and Scientific Awareness of Physical Activity

The harm of inadequate physical activity and the necessity of daily physical activity should be emphasized to people in all life cycle stages through various channels. There should be an emphasis on the concept of "prevention is better than cure, maintenance is better than prevention", which will promote the concept of "exercise is medicine". Creating a general environment for national fitness will also create a solid social foundation for the formulation and implementation of physical activity guidelines based on the life cycle concept.

### 5.3. Focus on Talent Cultivation and Improving Guideline Research and Development

The development of physical activity guidelines is a process of participation and consensus-building among professionals from multiple disciplines. While strengthening the training of reserve talents in various professional disciplines, there also needs to be a focus on the training of interdisciplinary talents. This should be done by providing relevant cross-disciplinary courses in medical schools or physical education colleges to train cross-disciplinary students in sports and medicine who are engaged in preparing physical activity guidelines at the theoretical and operational levels.

### 5.4. Mobilize All Sectors to Promote the Implementation of Physical Activity Guidelines

Mobilizing the efforts of all sectors of society, the government should provide policy and financial support at the macro level, professionals from the medical and health sectors

should provide theoretical support, sports and fitness instructors from the sports sector should provide technical guidance, and the urban construction sector should expand the site facilities needed by residents for physical activities. Moreover, organizations from all levels of society must cooperate to establish a service system of points, lines, and surfaces to cover all residents three-dimensionally and effectively promote the implementation of physical activity guidelines.

## 6. Conclusions

Governments in the USA, the UK, Australia, Japan, and Canada have issued physical activity guidelines for all life cycle stages, forming an effective system of policy tools to promote national physical activity. Among them, the relationship between physical activity and the health "dose–effect", the idea of "exercise is medicine", and the concept of 24-h healthy living have become the main theoretical basis for developing physical activity guidelines. The lack of daily physical activity in China has become a popular trend, and fitness has risen to become a national strategy to accelerate the realization of the strategic goal of Healthy China 2030, which was developed to promote, maintain, and protect people's health and improve health levels. Developing and popularizing physical activity guidelines to include all life cycle stages is an essential task and part of this initiative.

**Author Contributions:** F.L. and J.W.: study conception and design. J.W.: manuscript writing. F.L., J.W., L.R., Z.W., Y.S., T.X., L.W. and S.Y.: manuscript reviewing and modification. All authors have read and agreed to the published version of the manuscript.

**Funding:** This work was supported by the Key Program of Social Sciences of Jiangsu Province (Funder. Jiangsu Federation of Social Sciences; Grant No. 21WTA001); the Guangdong Scientific Project (Funder. Department of Science and Technology of Guangdong Province; Grant No. 2015A020219015); Major Science and Technology Demonstration Project of Jiangsu Science and Technology Department (Funder. Department of Science and Technology of Jiangsu Province; Grant No. BE2018752) and Major Scientific Research Project of Jiangsu Sports Bureau (Funder. Jiangsu Provincial Sports Bureau; Grant No. ST202102).

**Institutional Review Board Statement:** Not applicable.

**Informed Consent Statement:** Not applicable.

**Data Availability Statement:** Not applicable.

**Acknowledgments:** The authors greatly appreciate the support of Kui-Ting Gao, Jun-Feng Zhou, and the extraordinary research participants for their enthusiasm in participating in this project.

**Conflicts of Interest:** The authors declare that the research was conducted in the absence of any commercial or financial relationships that could be construed as a potential conflict of interest.

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
