# Peer review of "Development of Physical Activity Guidelines for a Healthy China Using the Life Cycle Concept: The Perspective of Policy Tools from Five Countries"

_sustainability, doi:10.3390/su141911956_

Round 1
Reviewer 1 Report
The creation of guidelines for both healthy and sick people represents an important factor that can lead to the improvement of human health, therefore, I positively evaluate the choice of topic.
1. add space recreational activities[2].
2. add space COVID-19 infection[3].
3. add source „In the post-epidemic era, Chinese people’s awareness of the importance of daily physical activity has also increased.“
4. add source „At the National Health and Wellness Conference in 2016, President Xi Jinping emphasized that China should focus on providing health and wellness services throughout the life cycle.“
5. change stylistic and interpretation „President Xi once again stressed that “The concept of life 8ycle health management should run through the whole process of urban planning, construction, and management.”“
6. add the precise name of all used database „One is through various databases, such as CNIK, PubMed, and Web of Science.“
7. add information about „how many articles was found“
8. better interpretation is needed „Correlation: To ensure the comprehensiveness of the analysis, as long as the document includes “physical activity volume,” “physical activity intensity,” “physical active- ity type,” or other related content about physical“
9. Why did the authors choose only five countries? Which is key for choosing country?
10 add space Table2. Encoding table of policy documents for physical activity guidelines in developed countries
11. I think, it is not a good idea to write about the conclusion in results sections „In conclusion, the five countries have developed a policy tool system comprising mainly supplied policies, supplemented by environmental and demand policy tools. However, they make comprehensive and effective use of the various policy tools. When selecting and assigning policy tools, the governments focus on the advantages and functions of the various policy tools and how they complement each other to form a reasonable system for promoting physical activity.“
Author Response
Q1: add space recreational activities [2].
Response: Thank you for this constructive suggestion. We have added“(e.g. jogging, cycling, dancing etc.) ”in the revised manuscript.
Q2. add space COVID-19 infection [3].n
Response: We thank the reviewer for the constructive suggestion and criticism. We have added“The latest research has demonstrated that physical activity plays an important role in countering COVID-19 infection through improving the body’s immunity, fighting against the new coronavirus infection and the recovery process” in the revised manuscript.
Q3. add source“In the post-epidemic era, Chinese people’s awareness of the importance of daily physical activity has also increased.”
Response: Thank you for pointing our negligence out. We have added the reference in the revised manuscript.
[4]Hou, G.D.; Sun, M.K.; Yang, X.J.; Yu, L.; Li, R. Study on the function, task and path of national fitness in the post-epidemic era. Liaoning Sport Science and Technology2021,43,1-5.
Q4. add source“At the National Health and Wellness Conference in 2016, President Xi Jinping emphasized that China should focus on providing health and wellness services throughout the life cycle.”
Response: Thank you for this constructive suggestion, based on the comments of other reviewers, we have deleted this paragraph.
Q5: change stylistic and interpretation“President Xi once again stressed that The concept of life cycle health management should run through the whole process of urban planning, construction, and management.”
Response: Thank you for this constructive suggestion, based on the comments of other reviewers, we have deleted the description.
Q6: add the precise name of all used database“One is through various databases, such as CNIK, PubMed, and Web of Science.”
Response: Thank you for your criticism and helpful comment. We have corrected it in the revised manuscript. “One is through various databases, such as China National Knowledge Infrastructure,PubMed, and Web of Science.”
Q7:add information about“how many articles was found”
Response: We thank the reviewer for the constructive suggestion and criticism. We added the“Following the screening principles above and due to incomplete statistics, 70 documents were chosen and numbered chronologically according to their publication date.”in the revised manuscript.
Q8:better interpretation is needed“Correlation: To ensure the comprehensiveness of the analysis, as long as the document includes “physical activity volume,” “physical activity intensity,” “physical activity type,” or other related content about physical”
Response: Thank you for yours the suggestion correct. we have made specific explanation in the revised manuscript in response to your questions. “Precision: To ensure the precision of the analysis, the documents we selected includes guidelines, principles, activity recommendations, and manuals, which involve the content of physical activity recommendations.”
Q9:Why did the authors choose only five countries? Which is key for choosing country?
Response: Thank you for your criticism and helpful comment. I am sorry to select physical activity guidelines of only 5 countries (the USA, Canada, Australia, the UK, and Japan) to analyze because of some limitations such as analytical method, language, access permissions and so on, which may face the issue of insufficient sample size. The key for choosing the five countries is that these five countries’ physical activity guidelines are more representative and have higher international recognition. In addition, some countries are facing the same social problems with China, such as aging and rising obesity rates. Therefore, the physical activity guidelines and policies can bring some enlightenments to China.
Q10:add space Table2. Encoding table of policy documents for physical activity guidelines in developed countries
Response:We appreciate this comment. We have recoded Table 2 following the reviewer’s suggestion. The specific coding of each dimension is as follows: in the X-dimension, supplied policy tools were coded as 01, environmental policy tools as 02, and demand policy tools as 03. In the Y-dimension, the USA was coded as US, Canada as CN, Australia as AU, the UK as UK, and Japan as JP. In the Z-dimension, fetal was coded as 01, childhood as 02, adolescence as 03, adult as 04, and older life as 05. If a document involved multiple stages, it was coded in both stages.
Table2 Encoding table of policy documents for physical activity guidelines in developed countries
|
Number |
Time |
Document |
Code |
|
1 |
1978 |
Recommendation on the Quality and Quantity of Physical Exercise to Promote and Maintain Adults' Health |
01-US-04 |
|
2 |
1985 |
Principles of Guiding Exercise During and after Pregnancy |
01-US-01 |
|
3 |
1988 |
Opinion Statement on the Physical Fitness of Children and Adolescents |
03-US-0203 |
|
4 |
1990 |
Recommendation on the Quality and Quantity of Physical Exercise to Promote and Maintain Adults' Health |
01-US-04 |
|
5 |
1990 |
Healthy Citizen 2000 |
02-US-01:05 |
|
6 |
1994 |
Exercise Guidelines During and after Pregnancy |
01-US-01 |
|
7 |
1995 |
Towards the Future-National Sports Standards: Content and Evaluation Guidelines |
01-US-02 |
|
8 |
1996 |
Physical Activity and Health |
03-US-0203 |
|
9 |
1997 |
Promoting Lifelong Physical Activity for Youth: A Guideline to Community and School Projects |
01-US-03 |
|
10 |
1998 |
Recommendation on the Quality and Quantity of Physical Exercise to Promote and Maintain Adults' Cardiopulmonary, Muscle Strength, and Flexibility |
01-US-04 |
|
11 |
1998 |
Physical Activity and Health Promotion for Adults: Evidence and Impact |
03-UK-04 |
|
12 |
1998 |
Physical Activity Guidelines for Active Living in Canada |
01-CN-04 |
|
13 |
1999 |
Health Activity Guidelines and Handbook for Canadian Elderly People |
01-AU-05 |
|
14 |
1999 |
Physical Activity Guidelines for Australian |
01-AU-04 |
|
15 |
2000 |
Basic Plan for Sports Rejuvenation |
02-JP-02:05 |
|
16 |
2000 |
Healthy Japan 21 |
02-JP-04:05 |
|
17 |
2000 |
American 3~5 Years Old Children's Sports Suitability Practice Program |
01-US-02 |
|
18 |
2000 |
Healthy Citizen 2010 |
02-US-01:05 |
|
19 |
2001 |
Five-Year Plan to Revitalize Early Childhood Education |
02-JP-02 |
|
20 |
2002 |
Benefits and Risks of Exercise During Pregnancy |
03-JP-01 |
|
21 |
2002 |
Exercise Guidelines during Pregnancy and Postpartum |
01-US-01 |
|
22 |
2002 |
A Positive Start: Sports Guidelines for Children from 0 to 5 Years Old |
01-US-02 |
|
23 |
2002 |
Physical Activity Guidelines for Canadian Children and Adolescents |
01-CN-0203 |
|
24 |
2002 |
Benefits and Risks of Exercise During Pregnancy |
03-AU-01 |
|
25 |
2003 |
Physical Activity Guidelines for School-age Children |
01-US-02 |
|
26 |
2003 |
Exercise Guidelines During Pregnancy and Postpartum |
01-CN-01 |
|
27 |
2004 |
At Least 5 Days a Week: About Physical Activity and Its Relationship with Health |
03-UK-0203 |
|
28 |
2006 |
Exercise Guidelines for Health Promotion 2006 |
01-JP-04 |
|
29 |
2006 |
Pregnancy Exercise Statement |
03-UK-01 |
|
30 |
2008 |
Physical Activity Guidelines for American |
01-US-01:05 |
|
31 |
2009 |
Appropriate Educational Practice Program for American Children Aged 0-8 |
01-US-02 |
|
32 |
2009 |
Australia's National Activity Recommendations for Seniors |
01-AU-05 |
|
33 |
2009 |
Physical Activity Guidelines for Australian 0-5 Years |
01-AU-02 |
|
34 |
2010 |
Healthy Citizen 2020 |
02-US-01:05 |
|
35 |
2010 |
Physical Activity Guidelines for Briton |
01-UK-01:05 |
|
36 |
2011 |
Children's Obesity Prevention Policy |
02-US-02 |
|
37 |
2011 |
Action Guidelines to Promote Youth Physical Activity |
01-US-03 |
|
38 |
2011 |
Sedentary Behavior Guidelines for Canada's Children and Adolescents |
01-CN-0203 |
|
39 |
2011 |
Physical Activity Guidelines for Canadian |
01-CN-01:05 |
|
40 |
2011 |
Physical Activity Guidelines (Under 5) |
01-UK-02 |
|
41 |
2011 |
Start Activity, Stay Active |
02-AU-0203 |
|
42 |
2011 |
Report of the Chief Medical Officer of the UK |
03-UK-01:05 |
|
43 |
2012 |
Basic Plans for Sports |
02-JP-02:05 |
|
44 |
2012 |
Healthy Japan 21 |
02-JP-0405 |
|
45 |
2012 |
Sports Guidelines for Children |
01-JP-02 |
|
46 |
2012 |
Physical Activity Improvement Strategy for American Adolescent |
02-US-03 |
|
47 |
2012 |
Physical Activity Guidelines for Canada's Children Aged 0-5 |
01-CN-02 |
|
48 |
2013 |
Guidelines for Toddler Sports |
01-JP-02 |
|
49 |
2013 |
Health Promotion Physical Activity Guidelines |
01-JP-02:05 |
|
50 |
2013 |
Sports During Pregnancy and Postpartum |
03-AU-01 |
|
51 |
2014 |
Physical Activity and Sedentary Guidelines for Australian |
01-AU-01:05 |
|
52 |
2014 |
Let Us Move: Reduce Sedentary Life and Live a Positive Life (18-24 years old) |
02-AU-04 |
|
53 |
2014 |
Sports During Pregnancy |
03-AU-01 |
|
54 |
2015 |
Exercise Guidelines During Pregnancy and Postpartum |
01-US-01 |
|
55 |
2015 |
Pregnancy and Exercise |
03-AU-01 |
|
56 |
2016 |
Canada's 24-hour Activity Guidelines for Children and Adolescents: Combining Physical Activity, Sedentary Behavior, and Sleep |
01-CN-0203 |
|
57 |
2016 |
Sports During Pregnancy and Postpartum |
03-AU-01 |
|
58 |
2017 |
Guide to Obstetrics and Gynecology Diagnosis and Treatment |
01-JP-01 |
|
59 |
2017 |
24-hour Exercise Guidelines for Canada's Toddlers |
01-CN-01 |
|
60 |
2017 |
Physical Activity Guidelines for Australian Children's (0-5 Years Old) |
01-AU-02 |
|
61 |
2018 |
Physical Activity Guidelines for American |
01-US-01:05 |
|
62 |
2018 |
Physical Activity Guidelines During Pregnancy in Canada |
01-CN-01 |
|
63 |
2018 |
Canada's Shared Vision of Increasing Physical Activity/Decreasing Sedentary Activity: Let us Exercise |
02-CN-01:05 |
|
64 |
2018 |
Exercise During Pregnancy and Postpartum |
03-AU-01 |
|
65 |
2019 |
Australian 24-hour Activity Guidelines for Children and Adolescents: Integrating Physical Activity, Sedentary Behavior and Sleep |
01-AU-0203 |
|
66 |
2019 |
Physical Activity Guidelines for Briton |
01-UK-01:05 |
|
67 |
2020 |
Exercise Guidelines During Pregnancy and Postpartum |
01-US-01 |
|
68 |
2020 |
Physical Activity Guidelines: Report of the Chief Medical Officer of the UK |
01-UK-01:05 |
|
69 |
2021 |
Physical Activity Guidelines for 18-64 Years Old Australian Adults |
01-UK-04 |
|
70 |
2021 |
Physical Activity and Exercise Guidelines During Pregnancy |
01-AU-01 |
Q11:I think, it is not a good idea to write about the conclusion in results sections “In conclusion, the five countries have developed a policy tool system comprising mainly supplied policies, supplemented by environmental and demand policy tools. However, they make comprehensive and effective use of the various policy tools. When selecting and assigning policy tools, the governments focus on the advantages and functions of the various policy tools and how they complement each other to form a reasonable system for promoting physical activity.”
Response:Thank you for this constructive suggestion. Following the reviewer’s suggestion, in the revised manuscript, we have deleted “In conclusion, the five countries have developed a policy tool system comprising mainly supplied policies, supplemented by environmental and demand policy tools. However, they make comprehensive and effective use of the various policy tools. When selecting and assigning policy tools, the governments focus on the advantages and functions of the various policy tools and how they complement each other to form a reasonable system for promoting physical activity.”
Q1: add space recreational activities [2].
Response: Thank you for this constructive suggestion. We have added“(e.g. jogging, cycling, dancing etc.) ”in the revised manuscript.
Q2. add space COVID-19 infection [3].n
Response: We thank the reviewer for the constructive suggestion and criticism. We have added“The latest research has demonstrated that physical activity plays an important role in countering COVID-19 infection through improving the body’s immunity, fighting against the new coronavirus infection and the recovery process” in the revised manuscript.
Q3. add source“In the post-epidemic era, Chinese people’s awareness of the importance of daily physical activity has also increased.”
Response: Thank you for pointing our negligence out. We have added the reference in the revised manuscript.
[4]Hou, G.D.; Sun, M.K.; Yang, X.J.; Yu, L.; Li, R. Study on the function, task and path of national fitness in the post-epidemic era. Liaoning Sport Science and Technology2021,43,1-5.
Q4. add source“At the National Health and Wellness Conference in 2016, President Xi Jinping emphasized that China should focus on providing health and wellness services throughout the life cycle.”
Response: Thank you for this constructive suggestion, based on the comments of other reviewers, we have deleted this paragraph.
Q5: change stylistic and interpretation“President Xi once again stressed that The concept of life cycle health management should run through the whole process of urban planning, construction, and management.”
Response: Thank you for this constructive suggestion, based on the comments of other reviewers, we have deleted the description.
Q6: add the precise name of all used database“One is through various databases, such as CNIK, PubMed, and Web of Science.”
Response: Thank you for your criticism and helpful comment. We have corrected it in the revised manuscript. “One is through various databases, such as China National Knowledge Infrastructure,PubMed, and Web of Science.”
Q7:add information about“how many articles was found”
Response: We thank the reviewer for the constructive suggestion and criticism. We added the“Following the screening principles above and due to incomplete statistics, 70 documents were chosen and numbered chronologically according to their publication date.”in the revised manuscript.
Q8:better interpretation is needed“Correlation: To ensure the comprehensiveness of the analysis, as long as the document includes “physical activity volume,” “physical activity intensity,” “physical activity type,” or other related content about physical”
Response: Thank you for yours the suggestion correct. we have made specific explanation in the revised manuscript in response to your questions. “Precision: To ensure the precision of the analysis, the documents we selected includes guidelines, principles, activity recommendations, and manuals, which involve the content of physical activity recommendations.”
Q9:Why did the authors choose only five countries? Which is key for choosing country?
Response: Thank you for your criticism and helpful comment. I am sorry to select physical activity guidelines of only 5 countries (the USA, Canada, Australia, the UK, and Japan) to analyze because of some limitations such as analytical method, language, access permissions and so on, which may face the issue of insufficient sample size. The key for choosing the five countries is that these five countries’ physical activity guidelines are more representative and have higher international recognition. In addition, some countries are facing the same social problems with China, such as aging and rising obesity rates. Therefore, the physical activity guidelines and policies can bring some enlightenments to China.
Q10:add space Table2. Encoding table of policy documents for physical activity guidelines in developed countries
Response:We appreciate this comment. We have recoded Table 2 following the reviewer’s suggestion. The specific coding of each dimension is as follows: in the X-dimension, supplied policy tools were coded as 01, environmental policy tools as 02, and demand policy tools as 03. In the Y-dimension, the USA was coded as US, Canada as CN, Australia as AU, the UK as UK, and Japan as JP. In the Z-dimension, fetal was coded as 01, childhood as 02, adolescence as 03, adult as 04, and older life as 05. If a document involved multiple stages, it was coded in both stages.
Table2 Encoding table of policy documents for physical activity guidelines in developed countries
|
Number |
Time |
Document |
Code |
|
1 |
1978 |
Recommendation on the Quality and Quantity of Physical Exercise to Promote and Maintain Adults' Health |
01-US-04 |
|
2 |
1985 |
Principles of Guiding Exercise During and after Pregnancy |
01-US-01 |
|
3 |
1988 |
Opinion Statement on the Physical Fitness of Children and Adolescents |
03-US-0203 |
|
4 |
1990 |
Recommendation on the Quality and Quantity of Physical Exercise to Promote and Maintain Adults' Health |
01-US-04 |
|
5 |
1990 |
Healthy Citizen 2000 |
02-US-01:05 |
|
6 |
1994 |
Exercise Guidelines During and after Pregnancy |
01-US-01 |
|
7 |
1995 |
Towards the Future-National Sports Standards: Content and Evaluation Guidelines |
01-US-02 |
|
8 |
1996 |
Physical Activity and Health |
03-US-0203 |
|
9 |
1997 |
Promoting Lifelong Physical Activity for Youth: A Guideline to Community and School Projects |
01-US-03 |
|
10 |
1998 |
Recommendation on the Quality and Quantity of Physical Exercise to Promote and Maintain Adults' Cardiopulmonary, Muscle Strength, and Flexibility |
01-US-04 |
|
11 |
1998 |
Physical Activity and Health Promotion for Adults: Evidence and Impact |
03-UK-04 |
|
12 |
1998 |
Physical Activity Guidelines for Active Living in Canada |
01-CN-04 |
|
13 |
1999 |
Health Activity Guidelines and Handbook for Canadian Elderly People |
01-AU-05 |
|
14 |
1999 |
Physical Activity Guidelines for Australian |
01-AU-04 |
|
15 |
2000 |
Basic Plan for Sports Rejuvenation |
02-JP-02:05 |
|
16 |
2000 |
Healthy Japan 21 |
02-JP-04:05 |
|
17 |
2000 |
American 3~5 Years Old Children's Sports Suitability Practice Program |
01-US-02 |
|
18 |
2000 |
Healthy Citizen 2010 |
02-US-01:05 |
|
19 |
2001 |
Five-Year Plan to Revitalize Early Childhood Education |
02-JP-02 |
|
20 |
2002 |
Benefits and Risks of Exercise During Pregnancy |
03-JP-01 |
|
21 |
2002 |
Exercise Guidelines during Pregnancy and Postpartum |
01-US-01 |
|
22 |
2002 |
A Positive Start: Sports Guidelines for Children from 0 to 5 Years Old |
01-US-02 |
|
23 |
2002 |
Physical Activity Guidelines for Canadian Children and Adolescents |
01-CN-0203 |
|
24 |
2002 |
Benefits and Risks of Exercise During Pregnancy |
03-AU-01 |
|
25 |
2003 |
Physical Activity Guidelines for School-age Children |
01-US-02 |
|
26 |
2003 |
Exercise Guidelines During Pregnancy and Postpartum |
01-CN-01 |
|
27 |
2004 |
At Least 5 Days a Week: About Physical Activity and Its Relationship with Health |
03-UK-0203 |
|
28 |
2006 |
Exercise Guidelines for Health Promotion 2006 |
01-JP-04 |
|
29 |
2006 |
Pregnancy Exercise Statement |
03-UK-01 |
|
30 |
2008 |
Physical Activity Guidelines for American |
01-US-01:05 |
|
31 |
2009 |
Appropriate Educational Practice Program for American Children Aged 0-8 |
01-US-02 |
|
32 |
2009 |
Australia's National Activity Recommendations for Seniors |
01-AU-05 |
|
33 |
2009 |
Physical Activity Guidelines for Australian 0-5 Years |
01-AU-02 |
|
34 |
2010 |
Healthy Citizen 2020 |
02-US-01:05 |
|
35 |
2010 |
Physical Activity Guidelines for Briton |
01-UK-01:05 |
|
36 |
2011 |
Children's Obesity Prevention Policy |
02-US-02 |
|
37 |
2011 |
Action Guidelines to Promote Youth Physical Activity |
01-US-03 |
|
38 |
2011 |
Sedentary Behavior Guidelines for Canada's Children and Adolescents |
01-CN-0203 |
|
39 |
2011 |
Physical Activity Guidelines for Canadian |
01-CN-01:05 |
|
40 |
2011 |
Physical Activity Guidelines (Under 5) |
01-UK-02 |
|
41 |
2011 |
Start Activity, Stay Active |
02-AU-0203 |
|
42 |
2011 |
Report of the Chief Medical Officer of the UK |
03-UK-01:05 |
|
43 |
2012 |
Basic Plans for Sports |
02-JP-02:05 |
|
44 |
2012 |
Healthy Japan 21 |
02-JP-0405 |
|
45 |
2012 |
Sports Guidelines for Children |
01-JP-02 |
|
46 |
2012 |
Physical Activity Improvement Strategy for American Adolescent |
02-US-03 |
|
47 |
2012 |
Physical Activity Guidelines for Canada's Children Aged 0-5 |
01-CN-02 |
|
48 |
2013 |
Guidelines for Toddler Sports |
01-JP-02 |
|
49 |
2013 |
Health Promotion Physical Activity Guidelines |
01-JP-02:05 |
|
50 |
2013 |
Sports During Pregnancy and Postpartum |
03-AU-01 |
|
51 |
2014 |
Physical Activity and Sedentary Guidelines for Australian |
01-AU-01:05 |
|
52 |
2014 |
Let Us Move: Reduce Sedentary Life and Live a Positive Life (18-24 years old) |
02-AU-04 |
|
53 |
2014 |
Sports During Pregnancy |
03-AU-01 |
|
54 |
2015 |
Exercise Guidelines During Pregnancy and Postpartum |
01-US-01 |
|
55 |
2015 |
Pregnancy and Exercise |
03-AU-01 |
|
56 |
2016 |
Canada's 24-hour Activity Guidelines for Children and Adolescents: Combining Physical Activity, Sedentary Behavior, and Sleep |
01-CN-0203 |
|
57 |
2016 |
Sports During Pregnancy and Postpartum |
03-AU-01 |
|
58 |
2017 |
Guide to Obstetrics and Gynecology Diagnosis and Treatment |
01-JP-01 |
|
59 |
2017 |
24-hour Exercise Guidelines for Canada's Toddlers |
01-CN-01 |
|
60 |
2017 |
Physical Activity Guidelines for Australian Children's (0-5 Years Old) |
01-AU-02 |
|
61 |
2018 |
Physical Activity Guidelines for American |
01-US-01:05 |
|
62 |
2018 |
Physical Activity Guidelines During Pregnancy in Canada |
01-CN-01 |
|
63 |
2018 |
Canada's Shared Vision of Increasing Physical Activity/Decreasing Sedentary Activity: Let us Exercise |
02-CN-01:05 |
|
64 |
2018 |
Exercise During Pregnancy and Postpartum |
03-AU-01 |
|
65 |
2019 |
Australian 24-hour Activity Guidelines for Children and Adolescents: Integrating Physical Activity, Sedentary Behavior and Sleep |
01-AU-0203 |
|
66 |
2019 |
Physical Activity Guidelines for Briton |
01-UK-01:05 |
|
67 |
2020 |
Exercise Guidelines During Pregnancy and Postpartum |
01-US-01 |
|
68 |
2020 |
Physical Activity Guidelines: Report of the Chief Medical Officer of the UK |
01-UK-01:05 |
|
69 |
2021 |
Physical Activity Guidelines for 18-64 Years Old Australian Adults |
01-UK-04 |
|
70 |
2021 |
Physical Activity and Exercise Guidelines During Pregnancy |
01-AU-01 |
Q11:I think, it is not a good idea to write about the conclusion in results sections “In conclusion, the five countries have developed a policy tool system comprising mainly supplied policies, supplemented by environmental and demand policy tools. However, they make comprehensive and effective use of the various policy tools. When selecting and assigning policy tools, the governments focus on the advantages and functions of the various policy tools and how they complement each other to form a reasonable system for promoting physical activity.”
Response:Thank you for this constructive suggestion. Following the reviewer’s suggestion, in the revised manuscript, we have deleted “In conclusion, the five countries have developed a policy tool system comprising mainly supplied policies, supplemented by environmental and demand policy tools. However, they make comprehensive and effective use of the various policy tools. When selecting and assigning policy tools, the governments focus on the advantages and functions of the various policy tools and how they complement each other to form a reasonable system for promoting physical activity.”
Reviewer 2 Report
Thank you for the opportunity to review the research entitled "Development of Physical Activity Guidelines for a Healthy China using the Life Cycle Concept: The Perspective of Policy Tools from Five Countries."
This manuscript reviewed the guidelines for physical activity on different life cycles and policies of five developed countries to provide essential insights into the Healthy China 2030 Planning Outline. Based on a well-planned search process, a group of experts selected and classified the guidelines/policies based on a three-dimensional framework (X = type of policy, Y = country, Z = life cycle) to provide an overview of the guidelines for physical activity and performed a critical analysis of the current situation in China. The topic is an important area for the journal audience and can strongly contribute to the development of a new, more up-to-date, and comprehensive physical activity guideline for the People's Republic of China.
Congratulations to the authors for outstanding work. I recommend the publication of the manuscript, and below are only small comments and thoughts to improve the manuscript.
• At first glance, it was challenging to understand the code in Table 2. I suggest eliminating the first number, which refers to the chronological order of the published document. This information is already in the first column. Second, I suggest encoding the dimension Y with the initials of each country (US, CN, AU, UK, and JP) to make it cleaner. For example, the code of the first document ("Recommendation on the Quality and Quantity of Physical Exercise to Promote and Maintain Adults' Health") would be: 01-US-04. For dimension Z that has numbers in sequence, I suggest: 01 to 05 or 01:05. As an example, the fifth document ("Healthy Citizen 2000") would be: 02-US-01:05;
• Review lines 267-270. I believe it would be: 150-300 min/week of moderate physical activity, 75-150 min/week of vigorous physical activity, or an equivalent combination of both;
• It would be important to highlight the importance of meeting the muscle-strengthening component (https://doi.org/10.1186/s40798-020-00271-w http://dx.doi.org/10.1136/bjsports-2022-105519) beyond the 150 min/week of MVPA, which was recently incorporated into physical activity guidelines (compared to aerobic). Contemplating the aerobic and muscle-strengthening components may differ in the population (https://doi.org/10.1371/journal.pone.0267277);
• What I missed the most was some discussion about the different domains of physical activity (occupational, domestic, transportation, and leisure time). High levels of work-related PA or leisure time can affect health differently. Can any insights be addressed based on the revised documents?
• I missed the authors discussing the recent review on PA accumulated in bouts of ≥10 min;
• This is just a personal comment (feel free to ignore). I am not a fan of the term "exercise is medicine” used for public health outreach; initially designed to inform and educate physicians and other healthcare professionals about exercise (lines 452-59). Term criticized by some authors: https://doi.org/10.1080/2159676X.2018.1476010; https://www.ncbi.nlm.nih.gov/pmc/articles/PMC7444006/.
To promote physical activity in the population, the WHO adopted the term "every movement counts." (https://cdn.who.int/media/images/default-source/health-topics/physical-activity/summary-infographic-guideline-on-physical-activity.jpg?sfvrsn=246f54b7_9); this is simple and a direct message of physical activity and sedentary behavior (also adopted by the American Heart Association: https://www.heart.org/en/healthy-living/fitness/fitness-basics/make-every-move-count-infographic).
Author Response
Q1:At first glance, it was challenging to understand the code in Table 2. I suggest eliminating the first number, which refers to the chronological order of the published document. This information is already in the first column. Second, I suggest encoding the dimension Y with the initials of each country (US, CN, AU, UK, and JP) to make it cleaner. For example, the code of the first document ("Recommendation on the Quality and Quantity of Physical Exercise to Promote and Maintain Adults' Health") would be: 01-US-04. For dimension Z that has numbers in sequence, I suggest: 01 to 05 or 01:05. As an example, the fifth document ("Healthy Citizen 2000") would be: 02-US-01:05;
Response:Thank you for this constructive suggestion. We have recoded Table 2 following the reviewer’s suggestion. In the Y-dimension, the USA was coded as US, Canada as CN, Australia as AU, the UK as UK, and Japan as JP.
Table2 Encoding table of policy documents for physical activity guidelines in developed countries
|
Number |
Time |
Document |
Code |
|
1 |
1978 |
Recommendation on the Quality and Quantity of Physical Exercise to Promote and Maintain Adults' Health |
01-US-04 |
|
2 |
1985 |
Principles of Guiding Exercise During and after Pregnancy |
01-US-01 |
|
3 |
1988 |
Opinion Statement on the Physical Fitness of Children and Adolescents |
03-US-0203 |
|
4 |
1990 |
Recommendation on the Quality and Quantity of Physical Exercise to Promote and Maintain Adults' Health |
01-US-04 |
|
5 |
1990 |
Healthy Citizen 2000 |
02-US-01:05 |
|
6 |
1994 |
Exercise Guidelines During and after Pregnancy |
01-US-01 |
|
7 |
1995 |
Towards the Future-National Sports Standards: Content and Evaluation Guidelines |
01-US-02 |
|
8 |
1996 |
Physical Activity and Health |
03-US-0203 |
|
9 |
1997 |
Promoting Lifelong Physical Activity for Youth: A Guideline to Community and School Projects |
01-US-03 |
|
10 |
1998 |
Recommendation on the Quality and Quantity of Physical Exercise to Promote and Maintain Adults' Cardiopulmonary, Muscle Strength, and Flexibility |
01-US-04 |
|
11 |
1998 |
Physical Activity and Health Promotion for Adults: Evidence and Impact |
03-UK-04 |
|
12 |
1998 |
Physical Activity Guidelines for Active Living in Canada |
01-CN-04 |
|
13 |
1999 |
Health Activity Guidelines and Handbook for Canadian Elderly People |
01-AU-05 |
|
14 |
1999 |
Physical Activity Guidelines for Australian |
01-AU-04 |
|
15 |
2000 |
Basic Plan for Sports Rejuvenation |
02-JP-02:05 |
|
16 |
2000 |
Healthy Japan 21 |
02-JP-04:05 |
|
17 |
2000 |
American 3~5 Years Old Children's Sports Suitability Practice Program |
01-US-02 |
|
18 |
2000 |
Healthy Citizen 2010 |
02-US-01:05 |
|
19 |
2001 |
Five-Year Plan to Revitalize Early Childhood Education |
02-JP-02 |
|
20 |
2002 |
Benefits and Risks of Exercise During Pregnancy |
03-JP-01 |
|
21 |
2002 |
Exercise Guidelines during Pregnancy and Postpartum |
01-US-01 |
|
22 |
2002 |
A Positive Start: Sports Guidelines for Children from 0 to 5 Years Old |
01-US-02 |
|
23 |
2002 |
Physical Activity Guidelines for Canadian Children and Adolescents |
01-CN-0203 |
|
24 |
2002 |
Benefits and Risks of Exercise During Pregnancy |
03-AU-01 |
|
25 |
2003 |
Physical Activity Guidelines for School-age Children |
01-US-02 |
|
26 |
2003 |
Exercise Guidelines During Pregnancy and Postpartum |
01-CN-01 |
|
27 |
2004 |
At Least 5 Days a Week: About Physical Activity and Its Relationship with Health |
03-UK-0203 |
|
28 |
2006 |
Exercise Guidelines for Health Promotion 2006 |
01-JP-04 |
|
29 |
2006 |
Pregnancy Exercise Statement |
03-UK-01 |
|
30 |
2008 |
Physical Activity Guidelines for American |
01-US-01:05 |
|
31 |
2009 |
Appropriate Educational Practice Program for American Children Aged 0-8 |
01-US-02 |
|
32 |
2009 |
Australia's National Activity Recommendations for Seniors |
01-AU-05 |
|
33 |
2009 |
Physical Activity Guidelines for Australian 0-5 Years |
01-AU-02 |
|
34 |
2010 |
Healthy Citizen 2020 |
02-US-01:05 |
|
35 |
2010 |
Physical Activity Guidelines for Briton |
01-UK-01:05 |
|
36 |
2011 |
Children's Obesity Prevention Policy |
02-US-02 |
|
37 |
2011 |
Action Guidelines to Promote Youth Physical Activity |
01-US-03 |
|
38 |
2011 |
Sedentary Behavior Guidelines for Canada's Children and Adolescents |
01-CN-0203 |
|
39 |
2011 |
Physical Activity Guidelines for Canadian |
01-CN-01:05 |
|
40 |
2011 |
Physical Activity Guidelines (Under 5) |
01-UK-02 |
|
41 |
2011 |
Start Activity, Stay Active |
02-AU-0203 |
|
42 |
2011 |
Report of the Chief Medical Officer of the UK |
03-UK-01:05 |
|
43 |
2012 |
Basic Plans for Sports |
02-JP-02:05 |
|
44 |
2012 |
Healthy Japan 21 |
02-JP-0405 |
|
45 |
2012 |
Sports Guidelines for Children |
01-JP-02 |
|
46 |
2012 |
Physical Activity Improvement Strategy for American Adolescent |
02-US-03 |
|
47 |
2012 |
Physical Activity Guidelines for Canada's Children Aged 0-5 |
01-CN-02 |
|
48 |
2013 |
Guidelines for Toddler Sports |
01-JP-02 |
|
49 |
2013 |
Health Promotion Physical Activity Guidelines |
01-JP-02:05 |
|
50 |
2013 |
Sports During Pregnancy and Postpartum |
03-AU-01 |
|
51 |
2014 |
Physical Activity and Sedentary Guidelines for Australian |
01-AU-01:05 |
|
52 |
2014 |
Let Us Move: Reduce Sedentary Life and Live a Positive Life (18-24 years old) |
02-AU-04 |
|
53 |
2014 |
Sports During Pregnancy |
03-AU-01 |
|
54 |
2015 |
Exercise Guidelines During Pregnancy and Postpartum |
01-US-01 |
|
55 |
2015 |
Pregnancy and Exercise |
03-AU-01 |
|
56 |
2016 |
Canada's 24-hour Activity Guidelines for Children and Adolescents: Combining Physical Activity, Sedentary Behavior, and Sleep |
01-CN-0203 |
|
57 |
2016 |
Sports During Pregnancy and Postpartum |
03-AU-01 |
|
58 |
2017 |
Guide to Obstetrics and Gynecology Diagnosis and Treatment |
01-JP-01 |
|
59 |
2017 |
24-hour Exercise Guidelines for Canada's Toddlers |
01-CN-01 |
|
60 |
2017 |
Physical Activity Guidelines for Australian Children's (0-5 Years Old) |
01-AU-02 |
|
61 |
2018 |
Physical Activity Guidelines for American |
01-US-01:05 |
|
62 |
2018 |
Physical Activity Guidelines During Pregnancy in Canada |
01-CN-01 |
|
63 |
2018 |
Canada's Shared Vision of Increasing Physical Activity/Decreasing Sedentary Activity: Let us Exercise |
02-CN-01:05 |
|
64 |
2018 |
Exercise During Pregnancy and Postpartum |
03-AU-01 |
|
65 |
2019 |
Australian 24-hour Activity Guidelines for Children and Adolescents: Integrating Physical Activity, Sedentary Behavior and Sleep |
01-AU-0203 |
|
66 |
2019 |
Physical Activity Guidelines for Briton |
01-UK-01:05 |
|
67 |
2020 |
Exercise Guidelines During Pregnancy and Postpartum |
01-US-01 |
|
68 |
2020 |
Physical Activity Guidelines: Report of the Chief Medical Officer of the UK |
01-UK-01:05 |
|
69 |
2021 |
Physical Activity Guidelines for 18-64 Years Old Australian Adults |
01-UK-04 |
|
70 |
2021 |
Physical Activity and Exercise Guidelines During Pregnancy |
01-AU-01 |
Q2: Review lines 267-270. I believe it would be: 150-300 min/week of moderate physical activity, 75-150 min/week of vigorous physical activity, or an equivalent combination of both.
Response: Thank you for your criticism and helpful comment. We have corrected it to “150–300 minutes of moderately vigorous physical activity per week, 75–150 minutes of vigorous physical activity or an equivalent combination of both is the current recommended amount of daily physical activity for adults in these countries’ physical activity guidelines.” in the revised manuscript. In addition, we also modified the corresponding content in Table 3 and marked in red.
Q3. It would be important to highlight the importance of meeting the muscle-strengthening component (https://doi.org/10.1186/s40798-020-00271-w http://dx.doi.org/10.1136/bjsports-2022-105519) beyond the 150 min/week of MVPA, which was recently incorporated into physical activity guidelines (compared to aerobic). Contemplating the aerobic and muscle-strengthening components may differ in the population.
( https://doi.org/10.1371/journal.pone.0267277 )
Response: Thank you for this constructive suggestion. In the revised manuscript, we have added the“It can be found from the above that muscle-strengthening exercise, which is a voluntary activity that includes the use of weight machines, hand-held weights or own body weight and has multiple and unique health benefits for people, plays an important role in physical activity guideline in all countries above. Studies have shown that compared to engaging in either the muscle-strengthening exercise guideline (≥ 2 sessions/week) or the aerobic MVPA guideline alone (≥ 150 min/week), the combination of both may be most beneficial for the prevention and/or management of multiple prevalent chronic health conditions.”
Q4. What I missed the most was some discussion about the different domains of physical activity (occupational, domestic, transportation, and leisure time). High levels of work-related PA or leisure time can affect health differently. Can any insights be addressed based on the revised documents?
Response: We thank the reviewer for the constructive suggestion and criticism. Following the reviewer’s suggestion, we have added the relevant content in the revised manuscript.
what’s more, there are four principal domains in which PA can be performed; leisure, work, transportation and domestic life, which have been shown to display independent associations with health outcomes. It is essential to recommend suitable exercise for different domains. For example, the Australia physical activity guidelines not only give out the standard of exercise duration, but also have different suggestions for different domains, such as “build activity into your day”, “active at work”, “active in doors” and so on.
Q5. I missed the authors discussing the recent review on PA accumulated in bouts of ≥10 min;
Response: Thank you for this constructive suggestion. We have added some discussion about PA accumulated in bouts of ≥10 min in the revised manuscript. “In addition, sedentary time is also highlighted in the guidelines and there are many practical tips to reduce sedentary behavior in daily life or at work, such as walking around when talking on your mobile phone or asking your boss for a “walk and talk” meeting rather than a sitting meeting. The consensus is reached that a small amount of activity is better than inactivity. Some studies have suggested that bouts of physical activity as short as 10 min are associated with similar health benefits to physical activity accumulated in longer bouts.”

Reviewer 3 Report
Thanks for the opportunity to read this article.
After reading, I have some comments that I present below.
I didn’t understand the article’s purpose when I reached the end. The title seems to indicate that how the Chinese FA recommendations were developed will be explained. However, only the evolution of the Chinese recommendations was presented.
Furthermore, the objective of talking about the recommendations of other countries is not perceived. It looks like a two-part job. This article looks like an academic work for a discipline.
Lines, 1-4. Title. If the article proposes to develop PA guidelines for China, why use perspectives from other countries?
Line 28. What does WHO mean? Any abbreviation must be explained the first time it is used. Then only the abbreviation can be used.
Lines 32-33. How do the authors know that “In the post-epidemic era, Chinese people’s awareness of the importance of daily physical activity has also increased?” It appears to be a sentence based on the authors’ perception. If this sentence is true, in which study was this observed?
Line 5. The previous sentence and this one are talking about the post-pandemic context. However, reference 5 is that 2018 (a date before the pandemic.
Lines 40-47. This entire part must be written again. This article has a strong political bent. Science must be neutral; therefore, the text must not be biased in presenting the vision of a head of state or a political party.
Lines 57-58. What does “relevant policies” mean?
Line 58. The concept of a developed country is controversial. It is preferable to use “economically developed countries”.
I understood the explanations given by the authors in looking for the documents. However, it was not explained in detail how the recommendations were researched. Imagine that I now wanted to do the same research. How should I do it to get the same results?
Line 103. How can there be recommendations for fetuses?
Author Response
Q1.Lines, 1-4. Title. If the article proposes to develop PA guidelines for China, why use perspectives from other countries?
Response: Thank you for your criticism and helpful comment. There are many reasons to use perspectives from other countries. First of all, the purpose of this article is to draw on the experiences of other countries in developing physical activity guidelines due to the Chinese physical activity guidelines are not mature. Secondly, these five countries’ physical activity guidelines are representative and have higher international recognition, which can bring some experience to China. In addition, some countries are facing the same social problems with China, such as aging and rising obesity rates. Therefore, the physical activity guidelines and policies can bring some enlightenments to China.
Q2.Line 28. What does WHO mean? Any abbreviation must be explained the first time it is used. Then only the abbreviation can be used.
Response: Thank you for pointing our negligence out. We have corrected it in the revised manuscript. “According to the WHO(World Health Organization, WHO)”
Q3.Lines 32-33. How do the authors know that “In the post-epidemic era, Chinese people’s awareness of the importance of daily physical activity has also increased?” It appears to be a sentence based on the authors’ perception. If this sentence is true, in which study was this observed?
Response: We thank the reviewer for the constructive suggestion and criticism. We have added the reference in the revised manuscript.
[4]Hou, G.D.; Sun, M.K.; Yang, X.J.; Yu, L.; Li, R. Study on the function, task and path of national fitness in the post-epidemic era. Liaoning Sport Science and Technology2021,43,1-5.
Q4.Line 5. The previous sentence and this one are talking about the post-pandemic context. However, reference 5 is that 2018 (a date before the pandemic.
Response: Thank you for pointing our negligence out. We have changed the reference 5 into
“Zhang, S.J.; Wang, H.P.; Liu, M.; Zhang, M.W. The values, missions and responsibilities of family sports for full-life cycle health in the Post-Epidemic Era. Journal of Nanjing Sport Institute 2020,19,12-16.”in the revised manuscript.
Q5.Lines 40-47. This entire part must be written again. This article has a strong political bent. Science must be neutral; therefore, the text must not be biased in presenting the vision of a head of state or a political party.
Response: Thank you for your criticism and helpful comment. We have deleted the entire part in the revised manuscript.
Q6.Lines 57-58. What does “relevant policies” mean?
Response: Thank you for yours the suggestion correct. The “relevant policies” means the policies related to the development of national physical activity guidelines, such as Healthy Citizen 2000 in the USA, Healthy Japan 21 in Japan and so on.
Q7.Line 58. The concept of a developed country is controversial. It is preferable to use “economically developed countries”
Response: Thank you for your criticism and helpful comment. We have changed the “developed countries” into “economically developed countries” and marked in red in the revised manuscript.
Q8.I understood the explanations given by the authors in looking for the documents. However, it was not explained in detail how the recommendations were researched. Imagine that I now wanted to do the same research. How should I do it to get the same results?
Response: Thank you for yours the suggestion correct. We analyzed the documents by constructing a three-dimensional framework whose X-dimension is about policy tools, Y-dimension is about country and Z-dimension is about the different stages of the life cycle and encoded the framework. The coding process of this paper went through three rounds with five participants. In the first round, after a preliminary reading of the selected policies, two people classified the kind of the policy tools to determine the coding of the X-dimension. After completion, the reviewers exchanged the encoding result with each other and invited a third person to make the second round to discuss the differences between the former result and determine the final version. The third round involved another two reviewers identifying the life cycle stages in every document we collected determine the code of Z-dimension. The coding table is not modified after everyone agrees and the author analyzed the results. The specific coding of each dimension is as follows: in the X-dimension, supplied policy tools were coded as 01, environmental policy tools as 02, and demand policy tools as 03. In the Y-dimension, the USA was coded as US, Canada as CN, Australia as AU, the UK as UK, and Japan as JP. In the Z-dimension, fetal was coded as 01, childhood as 02, adolescence as 03, adult as 04, and older life as 05. If a document involved multiple stages, it was coded in both stages. The encoding results are illustrated in Table 2.
Q9.Line 103. How can there be recommendations for fetuses?
Response: Thank you for your criticism and helpful comment. This article mainly use the concept of the life cycle, which divided the population into five stages: fetal, childhood, adolescence, adult, and the older, and make recommendations for the development of physical activity guidelines in different stages.
Round 2
Reviewer 1 Report
I demand the mentioned changes as sufficient. The authors managed to improve the quality of the manuscript.
Author Response
Thank you very much for your constructive and useful suggestions on improving the quality of this paper,which really helps us a lot. Best wishes for the success of your business.
Reviewer 3 Report
I stand by my previous decision.
The authors made some changes to the article, but the essence of the article is the same and therefore I maintain the same reservations presented above.
For example, I still don't see any sense in the presentation of recommendations from other countries.
Author Response
Thank you very much for your constructive and useful suggestions and we sincerely apologize for not being able to satisfy you with our revised manuscript.In order to improve the quality of our manuscripts, can you give more specific suggestions for revisions?Hope everything goes well with your work.